# Compute Where It Counts: Efficient Large Language Models via Learned Granular Sparsity

## Abstract

Sparsity-aware inference can dramatically shrink computation requirements by reducing the number of parameters, and thus FLOPs, used in each forward pass. Existing methods tend to be heuristic (zeroing activations below fixed thresholds, retaining top K activations, etc.). These methods do not directly optimize individual thresholds using gradient-based methods and experience sharp performance degradation beyond 50% sparsity. This paper describes CWIC (Compute Where it Counts), a method that makes sparsity thresholds learnable and contextual. CWIC is designed to enable conditional computation in models, allowing them to self-distribute sparsity across each weight matrix. It also enables models to allocate specific compute per token and sequence. This is augmented by "granular sparsity", a method that decomposes matrix columns into smaller "stripes" for more expressive sparsity patterns. We show that CWIC and granular sparsity can distill 2x (50% sparsity) to 6x (83% sparsity) more compute-efficient sparse models from Llama 3.2-1B and 3B. Sparsifying Llama3.2-1B to 66% sparsity with CWIC achieves a 15% increase in aggregate benchmark scores over doing the same with TEAL (Liu et al., 2025), the current SOTA activation sparsity technique. In this setting, CWIC speeds up inference wall clock times by 2.5x compared to Llama3.2-1B in both GPU and CPU settings. CWIC demonstrates promising scaling behavior. We study CWIC compute allocation patterns and find that little compute is dedicated to filler or replicated text and more to challenging benchmark questions.

## 1 Introduction

As large language models (LLMs) grow in parameter count to attain desired performance levels, their inference compute requirements grow in lockstep. Consumer devices cannot support large inference compute needs which now drive massive industry hardware expenses. This poses a bottleneck for many applications, especially agentic ones, that require high-speed, low-cost options for real-world deployment. Several methods have been proposed to improve LLM inference efficiency, including sparse Mixture of Experts (MoE) (Shazeer et al., 2017), quantization (Jin et al., 2024), ReLU-based sparsity (Mirzadeh et al., 2024), and activation sparsity (Lee et al., 2024; Liu et al., 2025; Zhang et al., 2025).

Quantization casts weights into lower-precision types. This tends to be less expressive than dynamic sparsity techniques as the same matrix is applied to all inputs. ReLU-based sparsity requires that the model use ReLU activations; however, modern models have adopted SwiGLU (Shazeer, 2020a) and other alternatives for superior performance. Activation sparsity methods such as CATS (Lee et al., 2024) zero out non-salient, activations lower than a threshold. Although activation sparsity can be input-dependent, no existing method directly learns activation thresholds. Furthermore, these prior works exhibit sharp performance degradation at sparsity exceeding 50%.

This work presents CWIC (Compute Where it Counts), a method to effectively train sparsity-aware models. CWIC is inspired by sparse autoencoders (Rajamanoharan et al., 2024) and uses straight-through-estimator (STE) (Bengio et al., 2013) to directly optimize activation thresholds (Lee et al., 2024).

This allows the model to (1) designate specific levels of sparsity to specific weight matrices and (2) dynamically allocate compute to specific tokens and sequences. It also gives us control over the desired sparsity level using a loss function.

To complement CWIC, we introduce *granular sparsity*, which partitions matrix columns into smaller *stripes*. This enables increased expressivity over the traditional approach of treating a matrix column as the "unit" of sparsity that can be turned "on" or "off" (Lee et al., 2024). Picking a stripe size of 512 retains the hardware acceleration associated with these methods.

We distill 1B and 3B models from the Llama 3.2 family (Grattafiori et al., 2024) under the CWIC framework into versions that use 2x-6x fewer Active Parameters (AP) per token. As in Figure 2, we find that CWIC outperforms TEAL (Liu et al., 2025) across all sparsity levels and exhibits a graceful performance tradeoff at higher sparsity levels (lower AP) while TEAL tended to collapse beyond 33% sparsity (3x reduction in AP). CWIC kernels match TEAL performance and CWIC scales as a sparsification method. From an inference lens, we find speedups proportional to AP reduction on both GPUs and CPUs (subsubsection 4.2.5).

Examining the FLOPs and Active Parameters assigned by CWIC models to benchmark tasks reveals that they naturally allocate less compute to "easier" tokens (such as role tokens, filler words, system prompt) and sequences (such as questions from ARC-Easy vs ARC-Challenge (Clark et al., 2018)).

## 2 RELATED WORK

Activation sparsity reduces computation requirements by zeroing small activations, allowing them to be skipped during matrix multiplications. Relufication (Mirzadeh et al., 2024) replaces pretrained LLM activation functions with ReLUs and inserts ReLUs elsewhere in the model to induce sparsity. After finetuning to recover performance, Relufication can reduce FLOP counts by up to 50% with almost no degradation. ProSparse (Song et al., 2025) builds on Relufication by adding an L1 penalty to ReLU activations to further increase sparsity.

GRIFFIN, Dong et al. (2024) exploits sequence-level activation similarity to define adaptive sequence-level sparsity patterns. Deja Vu (Liu et al., 2023) and ShadowLLM (Akhauri et al., 2024) predict sparsity on the fly by training small auxiliary MLPs to determine which weights matter to particular input sequences. Q-Sparse (Wang et al., 2024) discards all but the K largest channels of input vectors when computing linear layers. Q-sparse improves performance over compute-equivalent dense models and shows that sparsity degrades performance less on larger models.

Most similar to our work are CATS (Lee et al., 2024), TEAL (Liu et al., 2025), and R-SPARSE (Zhang et al., 2025), which zero all activations that are smaller than a *threshold*. Unlike our work, they do not directly optimize individual thresholds using gradient-based methods. CATS targets the same activation frequency with every threshold, TEAL optimizes thresholds using a greedy block-wise heuristic, and R-SPARSE uses a search algorithm in conjunction with singular value analysis. Note that these methods allow for dynamic sparsity (the sparse weights differ per input) unlike model pruning and N:M sparsity (Zhou et al., 2021) that fix the matrix for all inputs.

Mixture of Experts (MoE) activates certain sections of the neural network ("experts") to reduce active parameter counts. Unlike activation sparsity, MoE architectures typically use a learned routing mechanism to choose which experts to activate. Sparsely-Gated Mixture-of-Experts (Shazeer et al., 2017) proposed a gating network that incentivizes sparse, yet balanced, expert selection for language modeling and translation. DeepSeekMoE (Dai et al., 2024) demonstrated that combinatorial expert selection with more experts improves performance. Pham et al. (2024) demonstrate the selection of experts with the largest output magnitude is an effective routing strategy. Zhou et al. (2022) find that performance can improve if different tokens can receive different amounts of compute.

Sparse Autoencoders (SAEs) faithfully reconstruct the hidden state of a neural network while activating a small percentage of their features. Variants include top-k SAEs (Gao et al., 2024) that retain only k largest activations, and JumpReLU (Rajamanoharan et al., 2024) SAEs that retain only ReLU activations that exceed a learned threshold. JumpReLU makes sparsity learnable, and allows different numbers of features to activate for different examples. Ayonrinde (2024) showed reconstruction fidelity is improved when different numbers of features can be activated for different tokens.

## 3 METHODS

### 3.1 GRANULAR SPARSITY

Existing methods (Mirzadeh et al., 2024; Wang et al., 2024; Lee et al., 2024) exploit column sparsity (sometimes transposed and described as row sparsity) in matrix multiplications. When an input vector has a zero element, the computation for the corresponding matrix column can be skipped entirely. Columns are the units of conditional computation in these methods.

Inspired by the insight of DeepSeekMoE (Dai et al., 2024) that smaller and more configurable experts lead to better performance, we sought to create a more expressive sparsity mechanism. "Granular sparsity" partitions each column into a set of *stripes*, with each stripe activated independently. In GPU terms, one can think of a "stripe" as a portion of a column spanning a whole GEMV kernel tile across output dimension. This greatly increases the number of achievable sparsity configurations (Figure 1). We now describe Granular Matrix Multiplication where stripes are the unit of conditional computation.

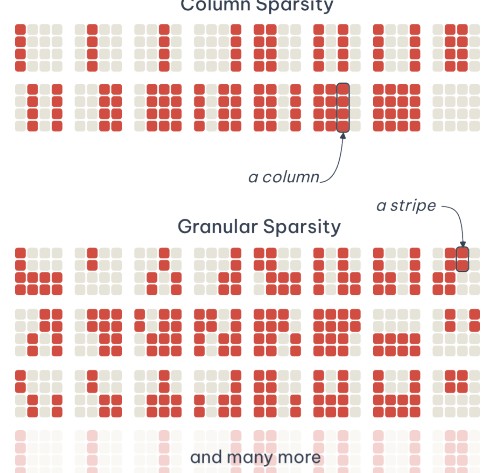

Figure 1: Given a 4x4 matrix, column sparsity results in 16 possible sparsity patterns. Partitioning each column into 2 individually activated stripes results in up to 256 configurations.

Let $\mathbf{W} \in \mathbb{R}^{m \times n}$, $\mathbf{y} \in \mathbb{R}^m$ and $\mathbf{x} \in \mathbb{R}^n$. The standard matrix-vector product can be rewritten as a weighted sum of the columns $\mathbf{W}_{:,i}$ of the matrix $W$.

$$\mathbf{y} := \mathbf{W}\mathbf{x} \in \mathbb{R}^m = \sum_{i=0}^{n-1} x_i \, \mathbf{W}_{:,i}. \tag{1}$$

Column-wise sparsity applies a binary mask $\mathbf{M} \in \{0,1\}^n$ to $\mathbf{x}$:

$$\mathbf{y} := \sum_{i=0}^{n-1} \mathbf{M}_i \, x_i \, \mathbf{W}_{:,i} = \sum_{i \in S_\mathbf{M}} x_i \, \mathbf{W}_{:,i}, \qquad S_\mathbf{M} := \{i \mid \mathbf{M}_i = 1\}. \tag{2}$$

In our method, we partition $\mathbf{y} \in \mathbb{R}^m$ and $\mathbf{W}_{:,i} \in \mathbb{R}^m$ into $k$ *stripes* of equal size $s = m/k$:

$$\mathbf{y} = \begin{bmatrix} \mathbf{y}_{0:s} \\ \vdots \\ \mathbf{y}_{(k-1)s:m} \end{bmatrix}, \quad \mathbf{w}_i = \begin{bmatrix} \mathbf{W}_{0:s,i} \\ \vdots \\ \mathbf{W}_{(k-1)s:m,i} \end{bmatrix}.$$

A binary *stripe mask* $\mathbf{G} \in \{0,1\}^{k \times n}$ selects which *stripes* of which columns are active. When $k = 1$, this reduces to standard column-wise sparsity. Using notation from 2:

$$\mathbf{y} = \begin{bmatrix} \sum_{i=0}^{n-1} \mathbf{G}_{0,i} \, x_i \, \mathbf{W}_{0:s,i} \\ \vdots \\ \sum_{i=0}^{n-1} \mathbf{G}_{k-1,i} \, x_i \, \mathbf{W}_{(k-1)s:m,i} \end{bmatrix} = \begin{bmatrix} \sum_{i \in S_{\mathbf{G_0}}} x_i \, \mathbf{W}_{0:s,i} \\ \vdots \\ \sum_{i \in S_{\mathbf{G_{k-1}}}} x_i \, \mathbf{W}_{(k-1)s:m,i} \end{bmatrix}, \quad S_{\mathbf{G_j}} := \{i \mid \mathbf{G}_{j,i} = 1\} \tag{3}$$

We represent this operation as Granular Matrix Multiplication (GMM):

$$\mathbf{y} := \mathrm{GMM}(\mathbf{x}, \mathbf{W}, \mathbf{G}) \tag{4}$$

### 3.1.1 SPARSITY THRESHOLDS

In contextual sparsity, we use a different mask G for each input vector x (Liu et al., 2023). We follow previous work (Lee et al., 2024; Liu et al., 2025) in using the magnitude of each element in $x$ to determine the mask. Specifically, we use a grid of thresholds $\theta \in \mathbb{R}_+^{k \times n}$ such that $\mathbf{G}_{s,i}$ is 1 if and only if $x_i$ has a magnitude of at least $\theta_{s,i}$. We write this threshold function $T(z,t)$ using the Heaviside step function $H(z)$.

$$T(z,t) := H(|z| - t), \qquad H(z) = \begin{cases} 1, & z \geq 0, \\ 0, & z < 0. \end{cases} \tag{5}$$

We define Sparse Granular Matrix Multiplication (SGMM):

$$\mathrm{SGMM}(\mathbf{x}, \mathbf{W}, \theta) := \mathrm{GMM}(\mathbf{x}, \mathbf{W}, \mathbf{G}) \qquad \text{where} \qquad \mathbf{G}_{s,i} = T(x_i, \theta_{s,i}) \tag{6}$$

### 3.1.2 LEARNING SPARSITY THRESHOLDS

Previous works that employ threshold-based contextual sparsity rely on heuristics (Liu et al., 2025) or search algorithms (Zhang et al., 2025) to determine $\theta$. We seek better optimization by directly learning the thresholds through backpropagation. Unfortunately, the step function $H(z)$ is not differentiable. We thus build on the ideas introduced for JumpReLU SAEs (Rajamanoharan et al., 2024) to construct a straight-through-estimator (STE) (Bengio et al., 2013) with a pseudo-derivative that approximates the function's true derivative. This pseudo-derivative is defined as follows, with $K$ representing a kernel function and $\epsilon$ representing a tunable bandwidth.

$$\frac{\partial}{\partial z} H(z) := \frac{1}{\epsilon} K\left(\frac{z}{\epsilon}\right) \tag{7}$$

For our kernel function $K$, we use the same rectangle function as JumpReLU.

$$K(z) := H\left(z - \frac{1}{2}\right) - H\left(z + \frac{1}{2}\right) \tag{8}$$

For a more detailed analysis of this gradient estimator, we refer readers to the JumpReLU paper (Rajamanoharan et al., 2024).

When calculating $\frac{\partial}{\partial x_i} \mathcal{G}_{s,i}$ or $\frac{\partial}{\partial \theta_{s,i}} \mathcal{G}_{s,i}$ we set the corresponding $\epsilon_i$ equal to the batch-wise standard deviation of $x_i$ (which does not receive gradients), scaled by a constant uniform hyperparameter $\alpha_\epsilon$.

$$\epsilon_i := \alpha_\epsilon \mathrm{std}(x_i)$$

We initialize $\theta$ to a small value at the start of training (see Appendix B). To keep $\theta$ positive, we set $\theta = \max(0, \theta_{\min})$ after every parameter update. We multiply the learning rate of $\theta$ by $\sqrt{n}$ so that their learning landscape is more similar in scale to existing weight parameters. Additionally, we scale them by the estimated standard deviation of each hidden channel as described in 3.4.1.

### 3.1.3 STRAIGHT-THROUGH ESTIMATION & THRESHOLD GATE GRADIENTS

Previous work (Wang et al., 2024) shows that sparse models can benefit from using an STE (Bengio et al., 2013) on the activation gradients during training. When back-propagating the AP loss gradients on the AP mask $T(x,t)$ introduced in 3.1.1, we allow gradients to flow to both $x$, and $t$. When back-propagating the gradients from $\mathcal{G}_{r,i} x_i = T(|x_i|, \theta_{r,i}) x_i$ we make two key changes. First, we take $x_i$' gradients straight through as if $x_i$ was unmasked. Second, we stop the gradients from flowing to $|x_i|$. To explain why the first part leads to the best results, we refer to the experiments in Q-Sparse (Wang et al., 2024) which show that the STE prevents vanishing gradients. The second part is needed because the scales of gradients going to $x_i$ through $|x_i|$ were dominating in practice due to the normalization step outlined in subsubsection 3.4.1. We also theorize that the STE reduces gradient variance when the sparsity masks are shifting frequently, but we leave this as speculation.

### 3.2 CONTROLLING SPARSITY

A key advantage of learnable thresholds is that we can directly control the sparsity of the model using a loss function. The number of active parameters (APs) to compute $\mathrm{SGMM}(\mathbf{x}, \mathbf{W}, \theta)$ can be

calculated based on the sparsity mask $\mathbf{G}$ and differentiated using the pseudo-derivative from section 3.1.2.

$$\text{APs}(\mathbf{x}, \mathbf{W}, \theta) := \frac{m}{k}||\mathbf{G}||_1 \tag{9}$$

We define $\text{APs}(B)$ as the number of active parameters required by the entire model to operate on a batch $B$, $\text{APs}_{\text{base}}(B)$ to represent the number of active parameters used by an equivalent dense model, and $\text{APs}_{\text{target}}(B)$ to represent the desired active parameter count. We then define the Active Parameter Reduction (APR) as the ratio between the base parameter count and the sparsely activated parameter count.

$$\text{APR}(B) := \frac{\text{APs}_{\text{base}}(B)}{\text{APs}(B)} \qquad \text{APR}_{\text{target}}(B) := \frac{\text{APs}_{\text{base}}(B)}{\text{APs}_{\text{target}}(B)} \tag{10}$$

We define our sparsity-controlling loss function using these quantities. This particular loss was chosen because it gives us control over the desired compute costs and stable performance during training.

$$\mathcal{L}_{\text{APs}} := \left(\min\left(\text{APR}\left(B\right) - \text{APR}_{\text{target}}\left(B\right), 0\right)\right)^2 \tag{11}$$

We found it very important to include a warm-up phase where $\text{APR}_{\text{target}}$ is incrementally increased. We use a linear schedule where $\text{APR}_{\text{target}}$ starts near 1 and ends at our desired sparsity level. This allows the model to smoothly and continuously adjust its weights in response to increased sparsity.

### 3.3 MLP Sparsity

Previous works have found that intermediate activations of the MLP blocks exhibit natural sparsity (Mirzadeh et al., 2024). To leverage this, we slightly modify our granular sparsity system in the MLP blocks.

The Llama 3 suite of models (Grattafiori et al., 2024) uses gated linear units (GLU) (Shazeer, 2020b). These are parameterized by a gate matrix $W_{\text{gate}} \in \mathbb{R}^{n \times d}$, an up matrix $W_{\text{up}} \in \mathbb{R}^{n \times d}$, and a down matrix $W_{\text{down}} \in \mathbb{R}^{d \times n}$.

$$y_{\text{GLU}} = W_{\text{down}}\left(W_{\text{up}}\boldsymbol{x} \odot \text{silu}(W_{\text{gate}}\boldsymbol{x})\right) \tag{12}$$

We compute $W_{\text{gate}}x$ using the standard GMM operation. Then, we use our learned threshold method to compute an activation sparsity mask $\mathcal{M} \in \{0, 1\}^d$ based on the magnitudes of $\text{silu}(W_{\text{gate}}x)$.

$$y_{\text{GLU, granular}} = W_{\text{down}}\left(W_{\text{up}} \odot a\right) \text{ where } a = \mathcal{M} \odot \text{silu}\left(\text{GMM}\left(x, W_{\text{gate}}, \mathcal{G}\right)\right) \tag{13}$$

When the operations rendered unnecessary by the mask are filtered out, we arrive at the AP count of this operation.

$$\text{APs}(\mathcal{G}_{\text{gate}}, \mathcal{M}) = \frac{d}{k}||\mathcal{G}_{\text{gate}}||_1 + 2n||\mathcal{M}||_1 \tag{14}$$

### 3.4 Model Training through Distillation

In this paper, we initialize the sparse model with the weights of a dense base model (Llama-3.2-1B and Llama-3.2-3B). To "sparsify" this dense model to desired levels, we use knowledge distillation (Hinton et al., 2015) where the teacher network is the dense base model and the student model is the sparsified model. For our distillation loss, over a sequence of length $T$, we use a sum of the forward (FKL) and reverse KL divergence (RKL) which we refer to as SKL, which has been shown to work better than either divergence individually (Wu et al., 2024). Total loss is a weighted combination of our distillation loss and our Active Parameter loss described in subsection 3.2:

$$\mathcal{L} = \text{SKL}(p_{\text{T}}(y_t|y_{<t}), p_{\text{S}}(y_t|y_{<t})) + \lambda_{\text{APs}}\mathcal{L}_{\text{APs}} \tag{15}$$

#### 3.4.1 Normalization and Mean Preservation

When initializing from a pre-trained network, we found that the batch-wise scales and offsets of $x_i$ values can vary throughout the network, which causes training instability if not addressed. To remedy this, we calculate the batch-wise mean $\bar{x} \in \mathbb{R}^n$ and standard deviation $\sigma(x) \in \mathbb{R}^n_+$.

Before applying the SGMM operation we scale $\theta_{s,i}$ by $\sigma(x)$, similar to the bandwidths in the previous section. We also subtract $\bar{x}$ from $x$ before sparsification, and add back $W\bar{x}$ (which can be baked into a simple bias term) afterwards. Note that neither $\bar{x}$ nor $\sigma(x)$ receive gradients.

$$y_{\text{stable}} := \text{SGMM}\left(x - \bar{x}, W; \theta \odot \sigma(x)\right) + W\bar{x} \tag{16}$$

Note that there is an strong theoretical reason to motivate the demeaning and rebiasing step: If we consider the mean an estimate of the mode of the channel distribution, then we are effectively ensuring the means and (roughly) modes of the output vector of the linear layer are preserved even when all stripes are turned off.

To decrease fluctuations, we track $\bar{x}$ and $\sigma(x)$ using an exponential moving average on a rolling basis determined by the hyperparameter $\beta_{\text{dist}}$. At inference time the running $\bar{x}$ and $\sigma(x)$ values from the last step of training are used.

## 4 EXPERIMENTS

### 4.1 SETUP

We tested our methods using teacher models Llama-3.2-1B [1] and Llama-3.2-3B [2]. We trained of a mix of open data defined in Appendix C. We used the AdamW optimizer (Loshchilov & Hutter, 2019). As a baseline, we applied TEAL (Liu et al., 2025) to the base model using default settings and the same dataset. Training was carried out over approximately 1B tokens for each experiment.

Distilling our 1B model required 52 hours on a single H100 GPU, which equates to only 0.015% of the 370K H100 hours originally used to train Llama-3.2-1B [3]. Our training implementation used only Python-level Pytorch operations, and we believe that a lower level implementation of the GMM operation, akin to Flash Attention (Dao et al., 2022), could accelerate training considerably. Checkpoints with APRs of less than 6x were taken from intermediate checkpoints corresponding to points in the target parameter reduction warm-up schedule (see subsection 3.2).

We aggregate model performance on standard LLM benchmarks including MMLU (Hendrycks et al., 2021), WinoGrande (Sakaguchi et al., 2019), ARC (Easy and Challenge) (Clark et al., 2018), HellaSwag (Zellers et al., 2019), PIQA (Bisk et al., 2019) and OpenBookQA (Mihaylov et al., 2018).

### 4.2 RESULTS

#### 4.2.1 CWIC OUTPERFORMS TEAL AND R-SPARSE

We find that CWIC outperforms TEAL (Liu et al., 2025) and R-SPARSE (Zhang et al., 2025) baselines across all sparsity levels when sparsifying both Llama3.2-1B and Llama3.2-3B (see Figure 2 and Table 1). While CWIC, TEAL and R-Sparse sparsified models exhibit performance drops compared to the dense teacher model, the degradation curves look significantly different. CWIC shows a gradual performance tradeoff as sparsity levels are increased (lower AP). This is explained by the increasing KL divergence we observe between high APR models and the original model. In contrast, TEAL and R-Sparse tend to exhibit performance collapse beyond 33% sparsity (3x reduction in AP). As a result of the divergence in performance-parameter curves, we find that 66% sparsified Llama3.2-1B with CWIC achieves a 15% improvement on aggregate benchmark scores compared to TEAL/R-Sparse sparsified models. This performance gap between the baselines and CWIC grows as the sparsity levels increase. The full set of results can be found in subsection A.1.

#### 4.2.2 CWIC OUTPERFORMS DENSE MODELS WITH COMPARABLE ACTIVE PARAMETERS

Table 1 also compares our 3x APR models to dense AP-equivalent transformer models in their compute classes that have been trained from scratch. We carry out this comparison since if a 1B model at 50% sparsity underperforms a dense model of 500M parameters, sparsification is a less worthwhile endeavor. CWIC marginally outperforms dense models with comparable AP on the

---

[1] meta-llama/Llama-3.2-1B

[2] meta-llama/Llama-3.2-3B

[3] meta-llama/Llama-3.2-1B

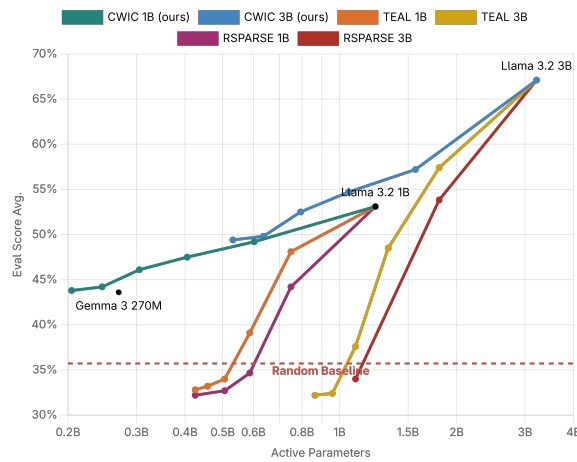

Figure 2: Average performance of CWIC and active-parameter equivalent models on standard LLM benchmarks (described in subsection 4.2). Each trend line shows performance of a base model at varying levels of sparsity (or active parameters) under CWIC, TEAL and R-Sparse. The rightmost point of each trendline corresponds to 0% sparsity.

benchmark aggregate. In Figure 2, a 3x AP reduction of Llama3.2-3B using TEAL underperforms Llama3.2-1B (a comparable AP model) by roughly 16%. In contrast, doing the same with CWIC outperforms Llama3.2-1B by 1.6%.

| Model Type (Active Params) | ALL | MMLU | WG* | Arc-C | HS* | Arc-E | OBQA* | PIQA |
|---|---|---|---|---|---|---|---|---|
| Llama3B RSPARSE 2.92x | 34.0 | 24.4 | 51.2 | 23.3 | 26.8 | 29.2 | 25.6 | 57.7 |
| Llama3B TEAL 2.92x | 37.6 | 23.2 | 50.1 | 26.6 | 34.5 | 41.7 | 26.0 | 61.3 |
| Llama3B CWIC 3.05x (1053M) | **54.7** | 38.3 | **58.3** | **39.5** | **64.3** | **69.4** | **39.4** | 73.9 |
| Llama1B (1236M) | 53.1 | **38.6** | 57.9 | 36.9 | 64.2 | 61.7 | 37.2 | **74.9** |
| Llama1B RSPARSE 2.91x | 32.2 | 23.2 | 49.6 | 24.7 | 25.4 | 26.6 | 26.0 | 50.0 |
| Llama1B TEAL 2.91x | 32.8 | 23.0 | 49.4 | 25.1 | 26.0 | 26.6 | 29.8 | 50.1 |
| Llama1B CWIC 3.05x (245M) | **44.2** | **25.2** | 51.6 | **29.2** | **48.3** | 52.8 | **33.4** | 68.8 |
| Gemma-3-270M | 43.6 | 24.3 | **52.7** | 27.6 | 43.8 | **57.5** | 30.6 | **68.9** |

Table 1: Comparison of 3x APR models to compute-comparable models. WG, HS and OBQA are the WinoGrande, HellaSwag and OpenbookQA datasets respectively.

While TEAL generates activations on a select set of sequences, CWIC distills a teacher model. TEAL thus requires less than an hour while CWIC requires on the order of days. These timescales are modest compared to methods like QSparse (which are used over the entirety of training time) or training a smaller dense model from scratch. While sparsification-time is incurred once, inference is recurring. Since TEAL and CWIC offer similar speedups (see subsubsection 4.2.5), performance becomes the dominant differentiator.

### 4.2.3 CWIC Ablation Findings

Three ablations tested CWIC design choices on Llama-3.2-1B. First, we test the utility of "granular sparsity" by reverting to column sparsity instead of stripe sparsity. Next, test the need for MSE on the last hidden states in the distillation loss. And finally testing our STE design choice. At APR 3.05x on Llama-3.2-1B, we observe a 1.7% performance hit using column sparsity and a 1.6% performance drop with MSE on the last hidden states (see Table 2). The full set of ablation results are provided in Appendix A.2.

### 4.2.4 CWIC Scaling Laws

Given budget constraints, we sparsified 1B and 3B models. We observed CWIC versions of both outperform their respective TEAL sparsified and dense active-parameter-equivilaents across sparsity levels (full results in subsection A.1).

| Version | Step | Multiplier | ALL | MMLU | WG | Arc-C | HS | Arc-E | OBQA | PIQA |
|---------|------|------------|-----|------|-----|-------|-----|-------|------|------|
| Full CWIC | 6000 | 3.05x | **47.5** | **26.7** | **55.7** | **33.1** | **53.6** | **58.0** | **34.8** | **70.4** |
| No stripes | 6000 | 3.05x | 45.8 | 26.2 | 54.1 | 30.9 | 51.4 | 56.3 | 32.6 | 69.1 |
| No MSE | 6000 | 3.05x | 45.9 | 24.7 | 52.2 | 30.7 | 51.4 | 57.4 | 34.4 | 70.5 |
| No STE | 6000 | 3.05x | 42.4 | 25.1 | 52.6 | 27.7 | 45.1 | 49.6 | 31.2 | 65.4 |

Table 2: Model performance under CWIC ablations

Modern LLMs can have 10-100x more parameters. Q-Sparse Wang et al. (2024) studied the scaling laws of sparsely activated models and found that the performance gap between dense and sparsely-activated models diminishes as model size increases. TEAL (Liu et al., 2025) shows that at 65% sparsity, Llama-3-8B4 sees an average downstream task performance reduction of 22% while Llama-3-70B5 (70B parameters) only sees a performance reduction of 9%. Since our method and TEAL both calculate sparsity based on activation thresholds, we expect to see similar scaling trends where CWIC will work as well, if not better, at larger scales

### 4.2.5 WALL CLOCK SPEEDUPS

We implemented a triton kernel based on TEAL (Liu et al., 2025). Keeping the stripe size a multiple of the triton kernel tile dimension along the output dimension maintains the efficiency of TEAL's triton kernel. In the GPU setting we matched the TEAL procedure. For testing on CPU we implemented a Rust version and compared to OpenBlas, taking precautions to evict the matrix from cache before each timed operation. At 66% sparsity, CWIC kernels speed up inference by 2.5x in both GPU and CPU settings for Llama3.2-1B and 2.7x for LLama3.2-3B. We describe the Triton kernel in more depth in Appendix F.

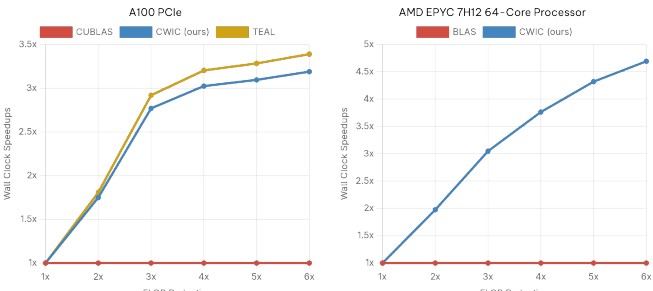

Figure 3: GPU and CPU speedups where hidden width: 4096, stripe size: 512

## 5 DISCUSSION

### 5.1 SPARSITY PATTERNS

The activation frequencies reveal several insights about the model's structure of circuits. First, we observe that some channels in the residual stream activate for almost every input across layers (Figure 7). These emerge early (100 steps), and persist for all of training. We believe that these channels may capture common knowledge, and serve a similar purpose to the shared experts used by DeepSeekMoE (Dai et al., 2024). An example of this behavior can be found in Appendix D. Ww also find patterns in the O attention matrix. Individual attention heads have consistently high/low activations across channels. We hypothesize that the model is learning to "prune" attention heads, similar to previous work that reduces compute cost by explicitly removing attention heads (Mugnaini et al., 2025).

Unlike the QKV, UP, GATE layers there are no patterns across layers as the input to O is not the residual stream. We have provided an example of this sparsity pattern in Appendix D.

We also observed stripes in the language modeling head corresponding to sections common vocabulary tokens tend to have higher activation frequencies than rare ones.

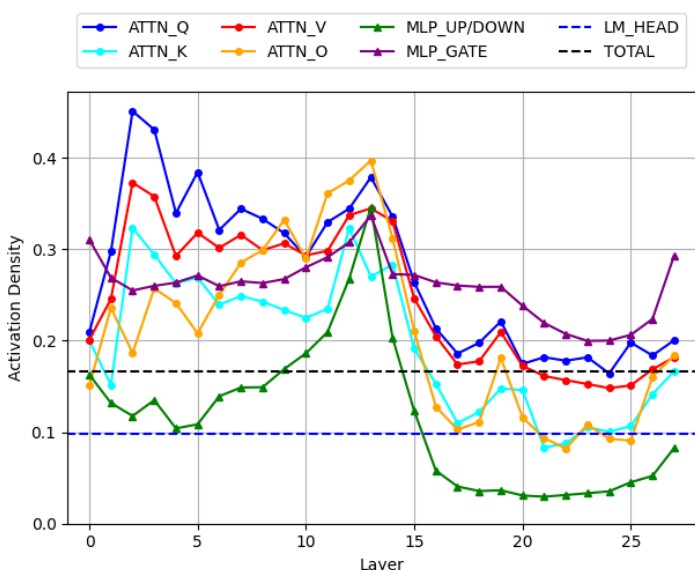

Figure 4: Activation frequency of different matrix types across layers from the 6x APR checkpoint of Llama-3.2-3B.

Finally, among the Q, K, and V attention matrices, activation of the V matrix is the most dense, followed by K and O (Figure 4). Note the two dips in activation frequencies (in layers 5-10) and (later layers 15-24) across matrices and spikes in early layers, middle layers 12-13 and final layer.

## 5.2 VARIABLE COMPUTE BUDGETS

Figure 5 uses text thickness to indicate compute allocated to different tokens in a prompt-response pair on a CWIC model based off Llama-3.2-1B-Instruct.

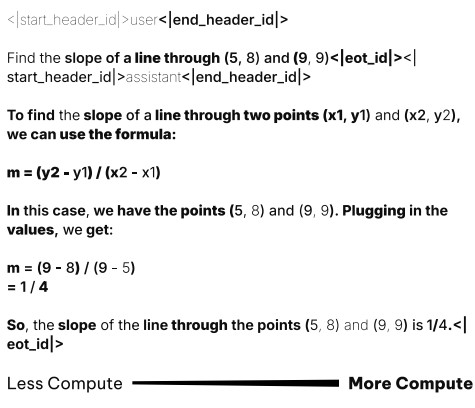

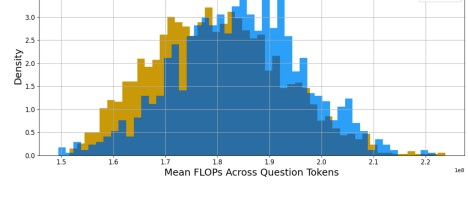

Figure 6: Active parameters allocated to each question of ARC-Easy and ARC-Challenge. We ignore the system prompt and only consider the question and option tokens. Active parameters are averaged per-question.

Figure 5: Active parameter count allocation of a 6x APR model across tokens.

We observe three common trends in compute allocation in general. Semantically quoting sections of the user prompt such as "(5,8) and (9,9)" in the response uses fewer active parameters. Punctuation, prepositions and filler words such as "the" and "and" are allocated lower budgets. Behavior on system prompt and system/user/assistant role tokens gets distilled into very few active parameters and thus receives a very low compute allocation. We provide more visualizations of variable compute allocation in Appendix E.

Sparsity thresholds enable different tokens and different sequences to use different amounts of compute. Figure 6 shows that the average Active parameters dedicated by the 6x APR model to running

ARC-Easy and ARC-Challenge benchmarks follow similar distributions, but with the Arc-E taking 5% less compute on average. A significant number of the ARC-E questions were allocated less compute than ARC-C questions. As evidenced by the scores of the 6x APR model (subsection A.1), ARC-Easy questions are indeed easier for the model!

## 6 FUTURE WORK

Our striping method groups output channels based on their order. However, outside of attention heads, there is no guarantee that adjacent channels are functionally similar. When initializing a sparse model from a pretrained one, it may be beneficial to reorder channels to form semantic groupings. This idea has seen success in mixture-of-expert conversions (Elazar & Taylor, 2022).

We observed that the KL divergence between the base and APR models remains stable between 2x-4.5x APR. It curves up beyond 4.5x APR. Performance improvements could be realized by modifying the APR warm-up to use a non-linear schedule.

We believe that several optimizations can be made to the GPU CWIC kernel to unlock further speed gains from Active Parameter reductions. Finally, we observed that longer training in tokens improves quality (as seen with the continued 6x training). It is natural to ask how far can this be pushed, and what are the scaling laws.

## LIMITATIONS

The primary limitation of our work concerns the scale of our experiments. We did not train models larger than 3B parameters, and we did not train for longer than 1B tokens. We believe that significantly better benchmark performance could be achieved with more training. Previous work (Wang et al., 2024) has also indicated that sparsity leads to less performance degradation at larger scales, so our method may be more suitable for larger models than those tested here. Furthermore, we only tested our method on transformer architectures and language modeling. The application to other models such as vision transformers (Dosovitskiy et al., 2021) has not been explored.

## REPRODUCIBILITY STATEMENT

We are committed to publishing reproducible research. We will open source our code (codebase already ready and version tracked on GitHub) and training data (already uploaded to HuggingFace). We will also be releasing model checkpoints for the research community to use and test out of the box (already uploaded to HuggingFace). We will link these resources in the paper after the review session during which anonymity is required. As a starting measure, we have provided all hyperparameter settings in Appendix B and the training data recipe in Appendix C.

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

# A EVALUATIONS RESULTS

## A.1 EXTENDED PERFORMANCE TABULATION

| Model | Technique | Step | Multiplier | all | mmlu | wg | arc:c | HS | arc:e | obqa | piqa |
|---|---|---|---|---|---|---|---|---|---|---|---|
| Gemma270M | — | — | — | 43.6 | 24.3 | 52.7 | 27.6 | 43.8 | 57.5 | 30.6 | 68.9 |
| llama-3.2-1B | — | — | — | 53.1 | 38.6 | 57.9 | 36.9 | 64.2 | 61.7 | 37.2 | 74.9 |
| llama-3.2-1B | CWIC | 3000 | 2.05x | 49.2 | 27.7 | 56.0 | 33.5 | 58.4 | 60.2 | 35.8 | 72.9 |
| llama-3.2-1B | CWIC | 6000 | 3.05x | 47.5 | 26.7 | 55.7 | 33.1 | 53.6 | 58.0 | 34.8 | 70.4 |
| llama-3.2-1B | CWIC | 9000 | 4.05x | 46.1 | 26.4 | 53.3 | 30.2 | 50.3 | 56.9 | 36.2 | 69.2 |
| llama-3.2-1B | CWIC | 12000 | 5.05x | 44.2 | 25.2 | 51.6 | 29.2 | 48.3 | 52.8 | 33.4 | 68.8 |
| llama-3.2-1B | CWIC | 15000 | 6.05x | 43.8 | 24.9 | 52.2 | 30.6 | 46.6 | 53.4 | 30.8 | 68.1 |
| llama-3.2-1B | CWIC | 18000 | 6.05x | 44.5 | 26.0 | 53.4 | 31.0 | 48.0 | 53.9 | 31.4 | 68.0 |
| llama-3.2-3B | — | — | — | 61.7 | 55.2 | 65.1 | 46.3 | 74.1 | 72.1 | 40.8 | 78.1 |
| llama-3.2-3B | CWIC | 3000 | 2.05x | 57.2 | 47.4 | 60.1 | 42.2 | 68.3 | 69.1 | 38.0 | 75.6 |
| llama-3.2-3B | CWIC | 6000 | 3.05x | 54.7 | 38.3 | 58.3 | 39.5 | 64.3 | 69.4 | 39.4 | 73.9 |
| llama-3.2-3B | CWIC | 9000 | 4.05x | 52.5 | 32.3 | 57.0 | 38.6 | 61.5 | 67.1 | 37.0 | 74.3 |
| llama-3.2-3B | CWIC | 12000 | 5.05x | 49.8 | 28.8 | 53.6 | 35.2 | 59.1 | 62.6 | 37.4 | 71.7 |
| llama-3.2-3B | CWIC | 15000 | 6.05x | 49.4 | 27.6 | 53.8 | 34.3 | 57.7 | 62.2 | 38.2 | 72.0 |
| llama-3.2-1B | TEAL | — | 1.65x | 48.1 | 28.1 | 55.3 | 34.0 | 55.2 | 56.6 | 35.6 | 71.7 |
| llama-3.2-1B | TEAL | — | 2.11x | 39.1 | 24.1 | 51.3 | 24.7 | 39.0 | 43.0 | 30.0 | 61.9 |
| llama-3.2-1B | TEAL | — | 2.44x | 34.0 | 22.9 | 48.0 | 22.7 | 28.9 | 32.8 | 28.0 | 55.0 |
| llama-3.2-1B | TEAL | — | 2.70x | 33.2 | 22.9 | 48.5 | 24.9 | 26.3 | 29.0 | 28.4 | 52.0 |
| llama-3.2-1B | TEAL | — | 2.91x | 32.8 | 23.0 | 49.4 | 25.1 | 26.0 | 26.6 | 29.8 | 50.1 |
| llama-3.2-3B | TEAL | — | 1.78x | 57.4 | 48.3 | 58.6 | 42.4 | 69.8 | 69.1 | 38.4 | 75.6 |
| llama-3.2-3B | TEAL | — | 2.41x | 48.5 | 30.7 | 54.5 | 35.2 | 55.4 | 58.3 | 36.0 | 69.4 |
| llama-3.2-3B | TEAL | — | 2.92x | 37.6 | 23.2 | 50.1 | 26.6 | 34.5 | 41.7 | 26.0 | 61.3 |
| llama-3.2-3B | TEAL | — | 3.35x | 32.4 | 23.0 | 50.4 | 23.6 | 26.4 | 26.4 | 26.6 | 50.7 |
| llama-3.2-3B | TEAL | — | 3.71x | 32.2 | 23.1 | 49.4 | 24.1 | 26.4 | 26.6 | 24.6 | 51.5 |
| llama-3.2-1B | R-SPARSE | — | 1.65x | 44.2 | 26.8 | 51.9 | 28.7 | 45.8 | 55.8 | 32.8 | 67.8 |
| llama-3.2-1B | R-SPARSE | — | 2.11x | 34.6 | 26.0 | 50.0 | 20.5 | 30.4 | 34.8 | 24.0 | 56.96 |
| llama-3.2-1B | R-SPARSE | — | 2.44x | 32.7 | 24.5 | 50.2 | 21.8 | 26.1 | 27.8 | 25.6 | 53.0 |
| llama-3.2-1B | R-SPARSE | — | 2.91x | 32.2 | 23.2 | 49.6 | 24.7 | 25.4 | 26.6 | 26.0 | 50.0 |
| llama-3.2-3B | R-SPARSE | — | 1.78x | 53.8 | 40.6 | 55.2 | 39.8 | 61.1 | 71.6 | 36.6 | 71.9 |
| llama-3.2-3B | R-SPARSE | — | 2.92x | 34.0 | 24.4 | 51.2 | 23.3 | 26.8 | 29.2 | 25.6 | 57.7 |

Table 3: Model Performance Comparison (acc_norm) using `lighteval==0.10.0` (Habib et al., 2023)

## A.2   CWIC ABLATIONS PERFORMANCE

| Model | Technique | Step | Multiplier | all | mmlu | wg | arc:c | HS | arc:e | obqa | piqa |
|-------|-----------|------|------------|-----|------|-----|-------|-----|-------|------|------|
| llama-3.2-1B | No stripes | 3000 | 2.05x | 48.0 | 27.3 | 54.8 | 30.7 | 56.9 | 57.1 | 35.8 | 73.3 |
| llama-3.2-1B | No stripes | 6000 | 3.05x | 45.8 | 26.2 | 54.1 | 30.9 | 51.4 | 56.3 | 32.6 | 69.1 |
| llama-3.2-1B | No stripes | 9000 | 4.05x | 45.2 | 26.5 | 53.1 | 30.2 | 49.5 | 55.7 | 33.2 | 68.3 |
| llama-3.2-1B | No stripes | 12000 | 5.05x | 43.9 | 26.9 | 51.9 | 30.7 | 46.6 | 52.0 | 31.6 | 67.7 |
| llama-3.2-1B | No stripes | 15000 | 6.05x | 42.4 | 23.6 | 51.4 | 28.7 | 44.6 | 50.0 | 31.6 | 66.7 |
| llama-3.2-1B | No MSE | 3000 | 2.05x | 48.3 | 26.8 | 54.4 | 34.0 | 56.3 | 59.8 | 35.2 | 71.8 |
| llama-3.2-1B | No MSE | 6000 | 3.05x | 45.9 | 24.7 | 52.2 | 30.7 | 51.4 | 57.4 | 34.4 | 70.5 |
| llama-3.2-1B | No MSE | 9000 | 4.05x | 45.6 | 25.8 | 53.5 | 29.8 | 48.9 | 57.2 | 35.8 | 68.1 |
| llama-3.2-1B | No MSE | 12000 | 5.05x | 45.0 | 26.3 | 56.3 | 29.2 | 46.6 | 53.8 | 34.0 | 68.8 |
| llama-3.2-1B | No MSE | 15000 | 6.05x | 44.4 | 24.9 | 53.2 | 29.7 | 46.5 | 55.1 | 33.4 | 68.2 |
| llama-3.2-1B | No STE | 3000 | 2.05x | 48.0 | 27.6 | 55.0 | 32.3 | 56 | 56.8 | 35.7 | 72.5 |
| llama-3.2-1B | No STE | 6000 | 3.05x | 42.4 | 25.1 | 52.6 | 27.7 | 45.1 | 49.6 | 31.2 | 65.4 |

Table 4: Model performance under CWIC ablations

# B   HYPERPARAMETERS

## B.1   SETTINGS

| Setting | 1B | 1B nostripes | 1B noMSE | 1B noSTE | 3B |
|---------|-----|--------------|----------|----------|-----|
| Base model | Llama-3.2-1B | Llama-3.2-1B | Llama-3.2-1B | Llama-3.2-1B | Llama-3.2-3B |
| Max sequence length | 1024 | 1024 | 1024 | 1024 | 1024 |
| Sequences per step | 64 | 64 | 64 | 64 | 64 |
| APR final | 6.05 | 6.05 | 6.05 | 6.05 | 6.05 |
| APR warmup length | 1.5e4 steps | – | – | – | – |
| LR schedule | 1e6 steps | – | – | – | – |
| LR warmup | 500 steps | – | – | – | – |
| LR max | 0.00005 | 0.00005 | 0.00005 | 0.00005 | 0.00003 |
| Optimizer | AdamW | AdamW | AdamW | AdamW | AdamW |
| Beta1 | 0.9 | 0.9 | 0.9 | 0.9 | 0.9 |
| Beta2 | 0.95 | 0.95 | 0.95 | 0.95 | 0.95 |
| Weight Decay | 0.01 | 0.01 | 0.01 | 0.01 | 0.01 |
| $\beta_{\text{dist}}$ | 0.99 | 0.99 | 0.99 | 0.99 | 0.99 |
| $\alpha_\epsilon$ | 0.1 | 0.1 | 0.1 | 0.1 | 0.1 |
| $\lambda_{\text{APs}}$ | 10.0 | 10.0 | 10.0 | 10.0 | 10.0 |
| stripe Size | 512 | FULL | 512 | 512 | 512 |

Table 5: Hyperparameter settings for the default model

The hyperparameters for our training run are presented in Table 4. Note that the default training mode had a APR target warmup, normalization, stripe size of 512, and x ste enabled.

## B.2   EFFECTS OF HYPERPARAMETERS

With respect to the bandwidth scale $\alpha_\epsilon$, performance was stable within a range of about 0.05 to 0.25. Values outside of this range caused significant training instability. The choice momentum parameter of running batch statistics $\beta_{\text{dist}}$ (range of 0.9 to 0.99) had very little impact on performance. When calculating pseudo-derivatives, we found that other kernel functions besides the rectangle kernel described in subsubsection 3.1.2 gave similar performance. Elongating the active parameter schedule over more train tokens in general improved the results in small scale experiments, but due to resource constraints we have not been able test these on full scale runs, say 2b or 4b token schedules.

## C  TRAINING DATA

Our training data consisted of sampled passages with sequence lengths of up to 1024 from the FineWeb-Edu (Lozhkov et al., 2024) dataset. The sampled mix will be published on huggingface. All data sequences were tokenized by the Llama-3.2 tokenizer, then filtered for a maximum total sequence length of 1024. We also packed shorter sequences together to increase training efficiency, and used attention masking to prevent interactions between packed sequences.

## D  OBSERVED SPARSITY PATTERNS

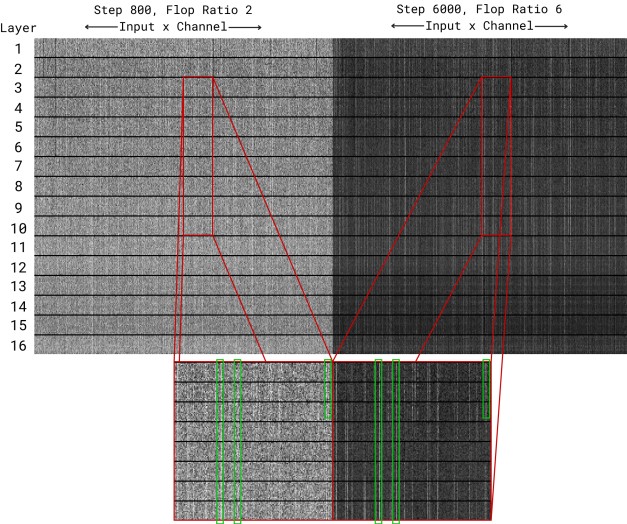

Figure 7: Activation frequencies of $W_{\text{gate}}$ at 2x and 6x APR. Rows correspond to weight stripes. Column intensity represents the frequency at which an input position passes the column threshold (darker column indicates lower frequency). The green boxes highlight how important features (brighter columns) emerge early and are magnified in relative importance over training.

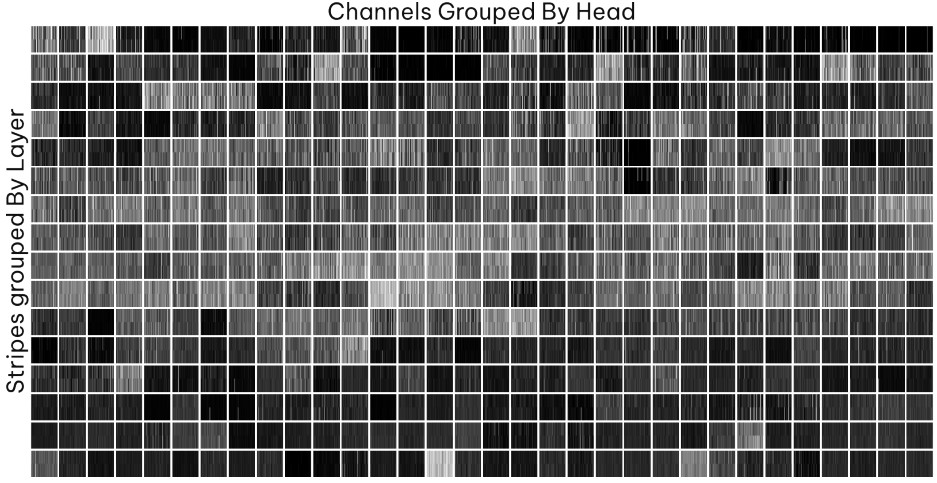

Figure 8: The sparsity levels *within* O heads (each white-outlined cell is a head) tend to be similar across channels. Sometimes entire heads will be always on or always off. Unlike the QKV, UP, GATE, there are no patterns across layers as the input to O is not the residual stream.

# E   OBSERVED COMPUTE ALLOCATION

<|begin_of_text|>Here is a simple rhyming poem with 4 lines about a lonely computer:
A screen that glows in silent night,
It waits for keys that never write.
No voices call, no hands appear,

<|begin_of_text|>Here is a list of 3 famous explorers from the early-modern period:
1. Christopher Columbus: Sailed across the Atlantic in 1492 and reached the Americas, initiating sustained European exploration there.
2. Ferdinand Magellan: Led the first expedition to circumnavigate the globe, proving the Earth could be fully traversed by sea.
3. Vasco da Gama: First European to reach India by sea via the Cape of Good Hope, opening direct maritime trade with Asia.

<|begin_of_text|>Question:
Bob has 5 baskets of apples. Each basket contains 3 apples. He gives 1 basket to Alice. How many apples does Bob have now?
Answer:
Step 1: Total apples initially = 5 baskets * 3 apples each = 15 apples.

0.0    0.1    0.2    0.3

Figure 9: Visualization of FLOPs allocated to different tokens during inference for three outut sequences.

# F   TRITON KERNEL DEVELOPMENT

The triton kernel used in this research is a derivative of the triton kernel developed by the authors of TEAL (Liu et al., 2025), and we would like to express our thanks for their baseline. Our main modification to the kernel was loading a threshold from memory per stripe instead of using a constant value. This naturally resulted in a slight wall clock overhead compared to TEAL at the same sparsity levels. Our triton kernel does not perform the debiasing and rebiasing, but future work could potentially fuse these for additional gains.

