# OpenReview forum: "Compute Where It Counts: Adaptive Compute Allocation for Large Language Models via Learned Granular Sparsity"
_ICLR.cc/2026/Conference — Submitted to ICLR 2026_

### Official Review · Reviewer_oPbm · 2025-11-01

**Soundness:** 2
**Presentation:** 2
**Contribution:** 3
**Rating:** 4
**Confidence:** 3

**Summary:**

CWIC (Compute Where it Counts) introduces a novel method for training sparse LLMs by making sparsity thresholds learnable parameters. The key contributions are: (1) learned contextual thresholds that are optimized using STEs, allowing models to dynamically allocate different amounts of compute to different tokens and weight matrices, and (2) granular sparsity that partitions matrix columns into smaller "stripes" for more expressive sparsity patterns.

**Strengths:**

I think the authors attempt to solve a really interesting research question of learning sparsity thresholds. I agree with the importance and limitations of existing works. I also found the idea around GMM quite interesting and intuitive. I found the

1) A really interesting approach to a principled gradient-based threshold learning - to directly optimize sparsity thresholds rather than using heuristics.
2) I quite liked the insightful analysis of learned sparsity patterns and discussion section.
3) Strong empirical results on variety of different benchmarks across multiple datasets.

**Weaknesses:**

Major Concerns

1) I am not sure about the claim - *STE  improves performance by removing the variance imparted on the grads when the values of G fluctuate...*: Using STE seems overly aggressive for the use-case? I think there needs to be more justification around this choice. Supporting experiments to compare that this is as a winning choice might also be helpful. Right now, this seems to more like a empirical tuning based selection?
- If this choice is derived from JumpReLU - their gradient estimator worked for SAE reconstruction doesn't automatically validate these additional modifications for multi-layer distillation. Current explanation is not sufficient for a convincing argument.

In general, I think section 3.3 is poorly written. Authors should re-word their ideas more clearly here.

2) No analysis of why related kernel choice is better than alternatives?

3) The psuedo-derivative bandwidth (lines 220-224): The interaction between adaptive bandwidth $(\epsilon_i = \alpha\epsilon · std(x_i))$ and input whitening (Section 3.4) is unclear - is std computed on whitened or raw values? How does this affect gradient scales?
- No comparison of adaptive vs. fixed bandwidth for pseudo-derivatives.

4) Experimental concern: The distillation data includes benchmark training sets (MMLU, ARC, WinoGrande) repeated 5×, which may inflate evaluation scores (refer: appendix C I believe). I think reporting scores on benchmarks not in training data would be a stronger claim.

5) Loss function: Authors need to explain how to interpret this. Right now, it looks asymmetrical: the loss only penalizes using too many parameters, not too few? In other words, it will keep on minimizing, i.e nothing stops the model from becoming 10× sparse if that happens to minimize distillation loss? It seems one would need careful tuning/warm starting to counter this or is there a implicit effect from distillation loss (or am I understanding this wrong)?

I strongly think the presentation wrt to the key ideas (section 3.2-3.6) need to be improved to make the author's work more accessible and make a convincing argument towards a lot of different empirical choices. Additionally, I found the concatenation operator to be a bit confusing (line 180-181), maybe there is a better way to mathematically denote it?

**Questions:**

NA

---

> ### Author Response · Authors · 2025-11-18
>
> Thank you for reviewing our work! We’re excited to see that you found the approach to threshold learning and takeaways from analysis of sparsity patterns interesting. We appreciate the attention to detail in your questions and general encouragement to improve the method section’s readability. It made us both realize we had missed mentioning an ablation and gave us the opportunity to more clearly motivate our choices. We have **rewritten chunks of Section 3 and grouped related subsections (e.g., “Sparsity Thresholds” now under “Granular Sparsity”) for a more logical flow of ideas**. We think this reads substantially better, thanks again for the feedback here!
>
> Addressing comments here pointwise for your further review:
>
> 1. On the use of STEs: From an empirical point of view, our ablations without STE did indeed underperform ones that used STE. We realised this was missing from our ablations table and have **now included it in our Results and Appendix for completeness**. As for the initial motivation, you are correct that it was inspired by JumpReLU. We agree that the estimator working for SAE reconstruction doesn't alone validate its utility here, so we turned to the empirical ablation for validation.
> 2. On kernel choice: We were only able to dedicate compute resources to running complete runs and ablations on a few design choices and had to be judicious with the settings we used. We did run mini-experiments to understand kernel behavior and guide our initial choice but did not feel they merited space in the paper as these experiments did not push the sparsity levels nearly as high as other experiments in this paper. We are certainly interested in alternatives to kernels and believe this is a topic for future research.
> 3. Thanks for the deep dive questions on our bandwidth choices! Clarifying here:
> - The bandwidth is always supposed to cover a set number of std. devs of the input distribution (in our case 0.1 stddevs). Since the distribution scales vary, to achieve this, the bandwidth must be “adaptive” and scale in line with the distribution.
> - We realized the “whitening” terminology was confusing and have **rephrased this in the Methods section**. The std dev is computed over raw values. We used “whitening” to refer to the process of standardizing but we realized using “standardization” would be clearer.
> - We have **clarified in Methods, a modification that was made to handle the effect of normalization** on gradient scales. Specifically, we stop the flow of gradients originating from the distillation losses to the |z| term in T (z, t) := H(|z| − t), but we don't detach them from the AP loss. We appreciate your feedback on this part of the paper and we will continue to edit it for more clarity. In particular, providing the motivation for our demeaning and rebiasing choice which stemmed from preserving the mean of the distribution when a channel is masked off by centering the mean at 0 before masking and adding it back.
> 4. Good catch, **thank you for alerting us**! The appendix was left over from an older version of this paper. The current benchmarks in this paper are from models that used **Fineweb-edu as the distillation set** and do not contain any benchmark data (to address exactly your concern). We have **updated the training data methodology and appendix  section** to be accurate now.
> 5. The impact of the loss term penalizing Active Parameters is indeed one directional. However, as you guessed, it is at odds with the distillation loss which itself would push the model to maintain density to maintain quality. During the distillation process we also only propagate gradients from the Active Parameter penalty if the active parameters used are above a scheduled threshold T(step). Over the process of distillation, we decrease this threshold from T_intial= 100% to T_final=16% of the model's initial parameters. Thus the active parameter penalty effectively pushes the AP down to the desired scheduled level but not further. We hope this clarifies the loss function construction!
>
> Finally, to make the notation easier to read, we **removed the concatenation operator and wrote out the matrix multiplication variants out in full** when introducing Granular Sparsity. Thanks again for the great feedback, looking forward to further thoughts or questions!

---

> > ### Author Response · Authors · 2025-11-28
> >
> > Since the discussion period is nearing its conclusion, we are writing to you with a gentle reminder to view and share your thoughts on our responses. If you have any additional questions or concerns, kindly let us know so we can run any additional experiments and resolve them before the discussion period concludes. If you feel our responses have satisfactorily addressed your concerns, we would be grateful for your reconsideration of this paper's score to show the same.
> >
> > We realize that this discussion period overlaps with the Thanksgiving holidays, so we want to express our gratitude for taking the time to share your thoughts here. Warm wishes for the upcoming holiday season.

---

### Official Review · Reviewer_dRcL · 2025-11-02

**Soundness:** 3
**Presentation:** 3
**Contribution:** 3
**Rating:** 6
**Confidence:** 3

**Summary:**

The paper suggests making sparsity thresholds both learnable and context-aware, with increased granularity. Experiments show these methods outperform alternatives.

**Strengths:**

1. The experiment results show the effectiveness of the method compared to alternatives in Pareto front trade-offs.
2. The authors made commendable efforts to achieve practical benefits through kernel implementations.
3. Several interesting insights emerged from the discussions.

**Weaknesses:**

Regarding technical details, the complexity of the proposed method is notable. Based on my understanding of Section 3, it appears to involve many hyperparameters.

This feedback is not intended as a reason to reject, but rather suggestions for improvement:
- The detailed discussion on "stripes" seems unnecessary. From a GEMM kernel perspective, this is simply (part of) a tile (?), and the proposed method essentially introduces sparsity within the tile level (or multiple tiles). This concept makes sense. I listed this in the weakness section, but it should be viewed as a recommendation for improvement. I suggest the authors emphasize this point. Similarly, from a hardware perspective, exploring 2D stripe sparsity, aligning stripes with the Tensor Core size, could be beneficial.

- Regarding Line 203, does the post hoc threshold update affect differentiability? It might be better to parameterize the threshold directly in log space, ensuring it remains positive.

- Section 3.3 seems poorly motivated. Have the authors considered common relaxations of binary variables, such as sigmoid or Gumbel-like methods?

**Questions:**

- Lines 45-46: "quantization ... tends to be less expressive than sparsity techniques..." Could the authors clarify this? I don't understand why quantization is considered less expressive, since I think of sparsity as a form of quantization to zero bits.

- How would batching be implemented?

- What are the conceptual differences between granular sparsity, N:M sparsity, and DeepSeek's Native Sparse Attention? (I understand they are used in different contexts, so I am asking about their conceptual similarities and differences).

- Section 3.8: this refers to Jensen–Shannon divergence.

- In the abstract, the method claims a wall clock speedup of 2.7x on GPU; is this compared to dense models (not TEAL)? Could the authors explain why their GPU implementation might be slower than TEAL? (Reasonable) speculations are okay with me.

- If I understand correctly, TEAL is somewhat training-free. Could the authors elaborate on how much slower their method is compared to TEAL?

---

> ### Author Response · Authors · 2025-11-17
>
> Thanks a lot for your review! We are excited that you noted the objective pareto performance improvements and intriguing qualitative observations of CWIC behavior. Your questions and pointers helped us revisit the work with fresh eyes, and we’ve incorporated many changes. Addressing your comments pointwise:
>
> 1. A stripe is “is simply part of a tile” – this is how we think about it too! However, in a previous iteration of this paper, this framing confused reviewers not as familiar with GPU programming. Out of concern that this description might not resonate widely, we have retained the current description. Please let us know if this is a sticking point though.
> 2. You are spot on about 2D sparsity. While we consider that future work, we do mention that “Keeping the stripe size a multiple of the triton kernel tile dimension along the output dimension maintains the efficiency of TEAL’s triton kernel”, as you suggested, in the “WALL CLOCK SPEEDUPS” section.
> 3. Regarding the post-hoc update: this update happens outside the autograd graph so it has no effect. We did consider log space but we thought this approach was cleaner since we start distillment with thresholds = 0 to exactly match the teacher model.
> 4. On relaxation methods, we did try sigmoid and gumbel among many other designs! We often found that learning would degrade early on in sparsification (first 100 steps). Based on those observations, we tried other methods until landing on our current kernel design. Given compute limits, we focused on the method that behaved well over full runs in favor of partial results from unstable early experiments. Hope this clarifies our thought process here.
>
> We enjoyed your questions. Addressing in order:
> 1. Your characterization of quantization as sparsity down to 0 bits is right. Here we specifically mean dynamic sparsity, which (1) doesn’t reduce model size and (2) allows different matrices per input. We have **clarified this in the text**!
> 2. The batching question is particularly insightful. Our technique inherits the limitations of TEAL, and the family of activation sparsity techniques which is that activations differ per time in the batch, making it challenging to share thresholds across a batch. At large batch sizes, we think there may be some clever techniques that group similar activation sequences together. We consider this future work at this time. It is worth noting that these methods are intended for edge inference on consumer hardware where the batch size is often 1 (we allude to this in the “Introduction” section motivating our technique).
> 3. Reviewer 1 asked a similar question about the difference between CWIC and structured model pruning (n:m sparsity) so we have **added it to our “Related Works section”**.
> - Weights used in each CWIC inference differs per token per sequence since it depends on the input. N:M sparsity and matrix factoring methods such as Monarch Matrices, make a one time reduction to the expressiveness and size of the weight matrix. This is more likely to break down on long tail behavior because neurons corresponding to long term behavior cannot conditionally activate. While pruning and N:M are often easier to accelerate with hardware due to less branching, dynamic sparsity (CATS, TEAL and CWIC) are more expressive.
> - DeepSeek’s Native Sparse Attention is an attention sparsity mechanism while CWIC is an activation sparsity method.  The direct goal of NSA is to sparsify the attention matrix by reducing which tokens are attended to. On the other hand, with CWIC-like methods,  the goal is to reduce the number of “non-zero” activations in hidden states. The benefits of these techniques (which could be used together) also differs. NSA is especially useful for long sequences since drop many key/query pairs. TEAL reduces the cost of the matrix mults across sequence lengths in the FFN layers. Hope this helps!
> 4. The distillation loss is not quite Jensen Shannon divergence. The JS divergence sums FKL and RKL conditioned on a _mixture_ of the two distributions.
> 5. Yes, the speedups on GPU/CPU compare CWIC models to the teacher/base dense model. These speedups are close (slightly less than) to those achieved by TEAL. We believe the source of this is that in our kernel, the threshold differs across different segments of a column while in TEAL, the threshold is constant across the entire column. As a result we have a few more memory loads than TEAL.
> 6. On sparsification time: at 50% sparsity, TEAL evaluates 1M samples in <1 hour; CWIC takes ~3.5 days. CWIC takes longer to apply to a model than TEAL, but both are modest compared to methods like QSparse (which are used over the entirety of training time) or even training a small dense model from scratch at which point the performance gap makes a big difference. **We note this in Section 4.2.2**.
>
> Overall, we enjoyed your review sharing many of our frameworks to solving in this space. We look forward to more discussion if you have follow-ups!

---

> > ### Author Response · Authors · 2025-11-28
> >
> > Since the discussion period is nearing its conclusion, we are writing to you with a gentle reminder to view and share your thoughts on our responses. If you have any additional questions or concerns, kindly let us know so we can run any additional experiments and resolve them before the discussion period concludes. If you feel our responses have satisfactorily addressed your concerns, we would be grateful for your reconsideration of this paper's score to show the same.
> >
> > We realize that this discussion period overlaps with the Thanksgiving holidays, so we want to express our gratitude for taking the time to share your thoughts here. Warm wishes for the upcoming holiday season.

---

### Official Review · Reviewer_AM9y · 2025-11-04

**Soundness:** 2
**Presentation:** 2
**Contribution:** 2
**Rating:** 2
**Confidence:** 4

**Summary:**

The paper proposes a new method to improve LLM efficiency. Computere Where It Counts (CWIC) introduces a granular sparsity pattern in the column vector of the matrix, in contrast to prior sparse activation methods, which mask the entire column.
The central claim of the paper is that using a granular sparsity pattern or stripes can improve the performance of the sparse model. The method achieves this by learning the threshold value which is then used by the gating function to mask the column vectors. Experiments are provided to compare the model with equivalent (same number of active parameters) dense models and method yeild wall-clock speedup by implementing Triton kernel.

**Strengths:**

* The use of granular sparsity is interesting and the ablation in 4.2.3 supports the idea of using granular sparsity.


* The authors also implemented the Triton kernel for the proposed method, which makes the method easy to deploy for real-world efficiency gains; however, the implementation details or code are not provided.


* Experiments in Table 1 support the claim that CWIC can match equivalent dense model performance; however, the comparison provided is limited.

* Analysis on the sparsity pattern (in section 5) is interesting and could be useful for the community.

**Weaknesses:**

* The methodology is not well-written and difficult to follow.

* The method is only compared against TEAL, more  sparse activation baselines should be added.

* The notations are not well-defined and method does not use the variables consistently. For e.g., in section 3.1, GMM functions takes $G$ as an argument but in section section 3.2, it takes $\theta$. Are G and $\theta$ same?

* Similarly, H(z) is first defined with only one argument (line 191) but in the next line it takes two arguments (x and t).

*  Minor: The statement about the scaling laws (section 4.2.4) is not convincing. It is possible that the CWIC may not follow the same trend as TEAL.

* It is not clear why and how distillation was used (section 3.8)? How do you initialize the teacher model? Is it a pre-trained model?

* More benchmark needs to be added to compare with other sparse activation baselines (table 1).

* Table 1 only shows the comparision between sparse and equivalent dense model. Comparison with the base model should be added to show the drop in performance is not significant.

* Comparison with structured sparsity/model pruning methods is missing and needs to be added to understand how CWIC is different from structured model pruning (n:m sparsity).

**Questions:**

1. The equation on page (section 3.1) does not seem to follow proper dimensions. Column vector v has $m$ dimensions but x has $n$ dimensions.

2. Why do we need normalization (section 3.4) separately? LayerNorm and RMSnorm already normalize the hidden features.

Suggestions:
1. Equations should be numbered and labelled, at least the main ones.


The paper overall needs a bit of rewriting, along with additional experiments/baselines.

---

> ### Author Response · Authors · 2025-11-17
> **Addressing Reviewer AM9y's Inputs and Questions**
>
> Thank you for your detailed feedback on the writing! We are glad that our introduction of granular sparsity and analysis of the qualitative performance of CWIC resonated with you.  Below, we address the areas for improvement and your questions.
>
> You had mentioned it would be helpful to share the kernel implementation. Our github make it easy to identify author identity which is why we planned to hold off until publication (as in our Reproducibility Statement). However **we realized we share the code more directly**! Please find it here: https://www.protectedtext.com/iclranon (password is "CWICICLR").
>
> 1. Your comments prompted us to rethink our presentation, and we ended up **rewriting/organizing the Methods and Results sections**.
> 2. We have **included R-Sparse (ICLR 2025)** as a competitive baseline (please see last comment in thread)!
> 2. Your note about notation inconsistencies was especially helpful. We **revised and reorganized the Methods section for clarity**. Key improvements include: Consistent notation (e.g., fixing inputs to GMM() and H()), Numbered equations and Grouping related subsections (e.g., “Sparsity Thresholds” now under “Granular Sparsity”).
> 3. It is true that larger models could further illustrate scaling, we demonstrate results across benchmarks on 1B and 3B models within our compute budget.
> 2. Re distillation, we realized that the training method was challenging to understand since it was referenced across the Introduction, Methods and Experiments section. Thank you for alerting us to this. We’ve now **consolidated the full distillation pipeline into a single “Model Training through Distillation” subsection**.
> 3. Addressing your questions about benchmarking, as far as we know TEAL (ICLR Spotlight 2025) is the SOTA activation sparsity technique. In the TEAL paper, the authors compare their technique solely to CATS (the prior SOTA). Therefore we felt it was sufficient to follow this precedent and compare the best available technique to validate CWIC. Is there another approach you would recommend comparing to as a baseline permitting our compute budget?
> 4. We **updated Table 1** in response to your comments. We aim to convey two findings in “CWIC outperforms TEAL” and “CWIC sparsification outperforms Active Parameter Equivalents.”
>   - For the former, **we compare (1) CWIC-sparsified models to the teacher and (2) CWIC to TEAL**. We moved the chart from the Introduction into the Results and clarified both sparsity–performance trends and the TEAL comparison.
>   - For the latter, we show that sparsifying a dense Llama-3B model down to a 1B-parameter sparse model still outperforms a dense Llama-1B model—essential to justify sparsification. For completeness, we have added a row for TEAL so that the performance jump from TEAL to CWIC is clear.
> 5. CWIC is unlike structured model pruning (n:m structured sparsity) in the same way that other activation sparsity methods are: the weights used or “pruned” are context dependent. The effective sparse set of weights used differs per token per sequence. On the other hand, n:m sparsity and matrix factoring methods such as Monarch Matrices, make a one time reduction to the expressiveness and size of the weight matrix. As the matrices are fixed, pruning/N:M sparsity is more likely to break down on long tail behavior because neurons corresponding to long term behavior cannot conditionally activate. Thus, while one-time structured sparsity (pruning and N:M) are often easier to accelerate with hardware due to less branching, dynamic sparsity (TEAL and CWIC) are more expressive. **We have added this to “Related Methods” for comprehensiveness.**
>
> Thank you for your questions! Point by point:
> 1. We took a second look at the equation in section 3.1 and confirmed its correctness as is. To aid readers, we have **added dimension breakdowns** for W, x and y. Since $\mathbf{y}  = \sum_{i=1}^n x_i\, \mathbf{w}_i$ where each column $\mathbf{w}_i \in \mathbb{R}^m$, $\mathbf{y}$ must lie in $\mathbb{R}^m$, following standard notation.
> 2. On normalization: RMSNorm in Llama includes a “post scale” per channel, resulting in a large variance in the scale of each channel's distribution. To alleviate the learning dynamics problem resulting from this, we track the distribution of activations and standardize them. This ensures the sensitivity in threshold learning dynamics are uniform across channels based on how much of the distribution they mask. Demeaning by $\mathbf{\mu_in}$ before the sparse matrix-vector multiplication and then rebiasing by $\mathbf{\mu_out}=\mathbf{W}\mathbf{\mu_in}$ allows the masked (stripe is off) behavior to fallback to preserving the mean of the output distribution.
>
> Finally, your suggestion to **label equations** was spot on. We have implemented this! Thanks again for helping us revisit our work with a fresh lens. We are grateful that this helped us communicate our ideas more effectively to our audience, thanks in large part to your feedback.

---

> > ### Comment · Reviewer_AM9y · 2025-11-18
> > **Reply to authors**
> >
> > Thank you for working on the presentation and providing additional explanations. I will re-read the methodology section again and will post follow-up questions, if required. Although your rebuttal answered most of my questions, I still have following concerns:
> >
> > * Use of distillation:
> >
> > The CWIC pipeline consists of two components: granular sparsity and distillation to recover the model's performance. TEAL, on the other hand, is training free. The central claim of the paper, in my opinion, is that having granular sparsity helps retain model performance (even at higher sparsity levels, where TEAL fails). Using both CWIC and distillation to recover model performance makes it difficult to evaluate whether CWIC is useful or if it is just distillation helping improve the model performance.
> >
> > * On normalization despite RMSNorms already appearing in the network:
> >
> > >  RMSNorm in Llama includes a “post scale” per channel, resulting in a large variance in the scale of each channel's distribution. To alleviate the learning dynamics problem resulting from this, we track the distribution of activations and standardize them
> >
> > Per-channel scaling in (batch norm/layernorm/rms norm) is important for the model to learn. I understand you want to get 'rid' of the scaling, for which you introduce an additional normalization step. Why not set the scaling factor in rms norm layer to 1 instead? I am not sure if I am missing something here but would like to know authors' intuition and explantion on this.

---

> > > ### Comment · Reviewer_AM9y · 2025-11-18
> > > **re:**
> > >
> > > Also, could you upload the code anonymously here (if you wish to)? https://anonymous.4open.science/
> > > This is a more popular way to upload the code; I personally have never used  https://www.protectedtext.com/, so bit reluctant to open the link in case it tracks IP location.

---

> ### Author Response · Authors · 2025-11-20
>
> Thanks for following up with us! We appreciate engaged reviewers.
>
> Operational: We've **uploaded the code here now**: https://anonymous.4open.science/r/anon-8BD5/cwic_triton/triton_torch.py
> You can access the entire codebase here in case that’s helpful. Thanks for pointing out this resource btw, will be helpful going forward.
>
> Moving on to your questions:
> 1. On the use of distillation: This is a good question that requires some nuanced consideration.
> - We think of CWIC as having three components – learnable thresholds, granular sparsity and distillation. It is straightforward to separate the granular sparsity component as we do in the ablations section. “No stripes” corresponds to using only the variable thresholds + distillation and treats a column as the unit of conditional sparsity. We see that on some benchmarks the ablation results in a 2-6% drop in performance suggesting that using granular sparsity does help.
> - Separating learnable thresholds and distillation is tricky. CWIC doesn’t use a smaller "base" model to distill since it simply sparsifies the larger model to reduce computation. Let us consider how we could distill into a smaller dense base model. This poses two complications. First, what could we use as the student initialization -- if we were to initialize with random values (unlike CWIC) and distill from scratch this would take far more tokens and time than CWIC simply because we aren’t able to “build upon” teacher weights. Second, how do we reach arbitrary FLOP reduction ratios -- in the case of CWIC, since the loss function uses the intended FLOP ratio target, we can mask a proportional number of params. However if we wanted a dense small model that was 4x sparser than LLama3B, we would have to create a custom dense 750M param architecture and independently decide how to size different layers.
> - The big benefit CWIC provides over pure distillation, is that (1) **we don't have to choose the student model size and architecture upfront** - the model allocates compute to the layers it determines most useful and (2) we get to reuse weights from the teacher. We considered how to create a pure distillation performance-compute curve, like we have in our paper. Since the research community hasn’t agreed on a standard distillation recipe, we realised we’d have to pick up smaller dense model architectures rather arbitrarily which would make extrapolating results challenging. If there are papers/huggingface repos you’ve come across that you think could hold an answer, we’d love to explore!
> - In addressing this, we came across https://pytorch.org/blog/llama-into-torchtune/ and found that the distilled 1B model underperforms CWIC on overlapping benchmarks. This can of course be attributed to a different recipe, but the fact remains that a convenient “standard” small student model with same tokenization doesn't exist for all flop ratios.
> - A speculative thought: If it were true that pure distillation of Llama 3B to 1B outperformed a from-scratch Llama 1B, that would likely be what was publicly released. We know that CWIC-ing Llama 3B to 33% outperforms dense Llama 1B. This has us suspecting that CWIC-ing Llama 3B to 33% will outperform purely distilling Llama3B to 1B. It seems that if CWIC-like eval improvements could be achieved with just distillation on < 500 million tokens, it would likely have already been done.
> 2. On normalization despite RMSNorms already appearing in the network: We have now **provided a theoretical justification in 3.4.1**. Setting the rms norm scales to 1 does not achieve our intended behavior for 2 reasons. First, the student would no longer match the teacher at initialization. Second, for an RMSNorm (not layernorm) the normalization uses RMS over the last hidden dimension across channels as the normalizing term for _all_ the channels. So the relative disparity in channel scales would not be removed. Hope this clarifies our process!

---

> > ### Author Response · Authors · 2025-11-28
> >
> > Since the discussion period is nearing its conclusion, we are writing to you with a gentle reminder to view and share your thoughts on our responses. If you have any additional questions or concerns, kindly let us know so we can run any additional experiments and resolve them before the discussion period concludes. If you feel our responses have satisfactorily addressed your concerns, we would be grateful for your reconsideration of this paper's score to show the same.
> >
> > We realize that this discussion period overlaps with the Thanksgiving holidays, so we want to express our gratitude for taking the time to share your thoughts here. Warm wishes for the upcoming holiday season.

---

> ### Author Response · Authors · 2025-12-03
>
> We looked for more relevant baselines and came across R-Sparse (ICLR 2025), another activation sparsity technique. We have **incorporated this additional baseline into our Results section**.
>
> Like TEAL, R-Sparse also reports outperforming CATS and GRIFFIN. Since R-Sparse and TEAL are contemporary methods (published at the same time), there is no comparison between them in the existing literature. We sparsified Llama 3B and 1B to varying levels with R-Sparse and evaluated on the lighteval v0.10.0 benchmark (like previous experiments).
>
> Across benchmarks, **R-Sparse slightly underperforms TEAL and significantly underperforms CWIC**. Interestingly, R-Sparse and TEAL follow very similar active parameter vs performance curves. Our conclusion remains that TEAL was the prior SOTA and based on benchmarks, **CWIC is the new SOTA** (result table provided below).
>
> Another observation: both R-Sparse and TEAL have an implied ceiling on sparsity levels since they do not sparsify the LMHead. Our best guess for why the authors made this design decision is that the LLMHead quickly collapsed under sparsity. **We note the ability to sparsify all weights stably as a unique strength of CWIC.**
>
>
> Comparison of 3x APR models to compute-comparable models:
>
> | **Model Type (Active Params)**      | **ALL** | **MMLU** | **WG\*** | **Arc-C** | **HS\*** | **Arc-E** | **OBQA\*** | **PIQA** |
> |-------------------------------------|:-------:|:--------:|:--------:|:---------:|:--------:|:---------:|:----------:|:--------:|
> | Llama3B  RSPARSE 2.92x              | 34.0 | 24.4 | 51.2 | 23.3 | 26.8 | 29.2 | 25.6 | 57.7 |
> | Llama3B  TEAL 2.92x                 | 37.6 | 23.2 | 50.1 | 26.6 | 34.5 | 41.7 | 26.0 | 61.3 |
> | Llama3B CWIC 3.05x (1053M)          | **54.7** | 38.3 | **58.3** | **39.5** | **64.3** | **69.4** | **39.4** | 73.9 |
> | Llama1B (1236M)                     | 53.1 | **38.6** | 57.9 | 36.9 | 64.2 | 61.7 | 37.2 | **74.9** |
> |-------------------------------------|--------|---------|---------|----------|---------|----------|-----------|---------|
> | Llama1B RSPARSE 2.91x               | 32.2 | 23.2 | 49.6 | 24.7 | 25.4 | 26.6 | 26.0 | 50.0 |
> | Llama1B TEAL 2.91x                  | 32.8 | 23.0 | 49.4 | 25.1 | 26.0 | 26.6 | 29.8 | 50.1 |
> | Llama1B CWIC 3.05x (245M)           | **44.2** | **25.2** | 51.6 | **29.2** | **48.3** | 52.8 | **33.4** | 68.8 |
> | Gemma-3-270M                        | 43.6 | 24.3 | **52.7** | 27.6 | 43.8 | **57.5** | 30.6 | **68.9** |
>
>
> WG, HS and OBQA are the WinoGrande, HellaSwag and OpenbookQA datasets respectively.

---

### Official Review · Reviewer_6kNm · 2025-11-17

**Soundness:** 3
**Presentation:** 2
**Contribution:** 3
**Rating:** 6
**Confidence:** 3

**Summary:**

The paper proposes CWIC, a sparsity-aware distillation method for large language models that learns activation thresholds in a context-dependent way and applies sparsity at a finer, stripe-level granularity inside linear layers. An explicit active-parameter (AP) loss is used to target a desired compute budget while distilling from dense Llama 3.2 models.

**Strengths:**

- The paper is well motivated. It addresses a very practical problem: reducing LLM inference cost while maintaining quality, in a way that can be applied directly to real Llama models.

- The core idea is clear: make sparsity thresholds learnable and context dependent, introduce stripe-level sparsity in the linear layers, and control everything with an explicit active-parameter budget.

- The main empirical message is easy to understand: CWIC achieves a better accuracy–compute trade-off than TEAL (SOTA model) over a range of sparsity levels, and the resulting sparse models slightly outperform dense models with comparable active parameters.

**Weaknesses:**

- The method requires substantially more training than TEAL, but the trade-off between extra training time and the observed gains is not analyzed in detail.

- There is no discussion of sensitivity to the many hyperparameters involved in the method (for example the APR schedule, the weight on the AP loss, stripe size, and normalization choices).

- Experiments are limited to Llama 3.2 1B and 3B models trained on roughly one billion tokens, so it is unclear how well the approach scales to larger models.

- Although the method is trained in a teacher–student (distillation) setting, it is not compared to standard distillation baselines such as smaller dense students trained with the same loss and similar compute.

**Questions:**

- How does the amount and composition of training data affect the benchmark scores and the accuracy–compute curves? For example, what happens if you train CWIC with more data, less data, or a different data mixture?

- Have you compared CWIC with non-sparse distillation baselines, such as dense student models distilled from the same teacher under the same objective?

- Can you quantify the training-time versus evaluation-quality trade-off relative to TEAL, which effectively incurs no additional training cost?

---

> ### Author Response · Authors · 2025-11-21
>
> Thank you for reviewing our work! We are glad the motivation, method and empirical performance improvements came across clearly. It is also clear that you grasped the core contributions of the paper, so these questions carry weight. We appreciate the thoughtful inquiry and areas of improvement which we address point wise for your further review. Some of your points were shared by our earlier reviewers and while we address them here, if you are interested you might find those threads engaging!
> 1. You are correct that since TEAL uses only activations it enjoys speed benefits. Empirically we find that at 50% sparsity, TEAL evaluates 1M samples in <1 hour; CWIC takes on the order of 3 days. CWIC takes longer to apply to a model than TEAL, but both are modest compared to methods like QSparse (which are used over the entirety of training time) or even training a smaller dense model from scratch. More importantly, the time to sparsify in one-and-done while inference is recurring. Since TEAL and CWIC offer similar speedups (see 4.2.5), performance now becomes the most important factor. Here, as you have noted, CWIC offers significant improvements over TEAL across sparsity levels. **For completeness, we have added this note to 4.2.2**.
> 2. Thanks for bringing up about hyperparameter sensitivity. Empirically we found that stripe size has the greatest impact. We thus **added an ablation “No stripes”** to demonstrate the performance drop when stripe size = column size instead of stripe size = column size/2. A slower schedule in general improved results, but we don't have any full-scale runs (one in which the model reached desired sparsity levels) on this. Distillation was not sensitive to the weight on the active parameters loss (unless there was an order of magnitude change.
> 3. Yes, we decided to use Llama 3B and 1B as base models to sparsify since they are generally accepted as “standard” small models. We ran over < 1B tokens to meet compute cost constraints and maintain reasonable iteration cycles. Since scaling is an important consideration for the community, we have explicitly **noted this in our Limitations section** for readers since our first draft.
> 4. On pure distillation – this is an important question. We did consider constructing this baseline. In doing so, we faced two complications. First, what do we use as the student initialization? If we were to initialize with random values (unlike CWIC) and distill from scratch this would take far more tokens and time than CWIC simply because we aren’t able to “build upon” teacher weights. Second, how do we reach arbitrary FLOP reduction ratios? In the case of CWIC, since the loss function uses the intended FLOP ratio target, we can mask a proportional number of params. However if we wanted a dense model that was 4x sparser than LLama3B, we would have to create a custom dense 750M param architecture and independently decide how to size different layers. Since the research community hasn’t agreed on a standard distillation recipe, we realised we’d have to pick smaller dense model architectures rather arbitrarily which would make extrapolating results challenging. As next best solution, **we picked comparable parameter SOTA models as baselines.**
>
> Some additional thoughts:
> - In https://pytorch.org/blog/llama-into-torchtune/ the distilled 1B model underperforms CWIC on overlapping benchmarks. This can of course be attributed to a different recipe, but the fact remains that a convenient “standard” small student model with same tokenization doesn't exist for all flop ratios.
> - A speculative thought: If it were true that pure distillation of Llama 3B to 1B outperformed a from-scratch Llama 1B, that would likely be what was publicly released. We know that CWIC-ing Llama 3B to 33% outperforms dense Llama 1B. This has us suspecting that CWIC-ing Llama 3B to 33% will outperform purely distilling Llama3B to 1B. It seems that if CWIC-like eval improvements could be achieved with just distillation on < 500 million tokens, it would likely have already been done.
>
>  This highlights one big benefit CWIC offers over pure distillation, that **(1) we don't have to choose the student model size and architecture upfront** - the model allocates compute to the layers it determines most useful and (2) we get to reuse weights from the teacher. We remain interested in clever ways to address this question so if there are papers/huggingface repos you’ve come across that you think could hold an answer, we’d love to explore!

---

> ### Author Response · Authors · 2025-11-21
>
> Moving on to your questions:
> 1. We used general text datasets like Fineweb for training. It is possible that picking a particular subset of data for an intended task (say summarization, financial knowledge, casual/slang text generation) could lead to better performance! We do have different runs using varying different data set mixtures (OpenHermes, Instruct text and Fineweb) but didn’t have any significant takeaways since the eval scores were all similar. So we decided to demonstrate the method using the simplest configuration of just one standard dataset (this also helped allay concerns of benchmark train sets being part of the distillation data in that variant)
> 2. The objective we use is a combination of distillation loss and active parameter reduction (APR). We might not be fully understanding your question, but the APR component would naturally make this incompatible with dense student models. As for dense baselines themselves, we shared our thought process above in (4).
> 3. We addressed the question of TEAL vs CWIC run times in (1) above! As mentioned there, we have also noted this point in our paper now.
>
> Thanks again for the probing questions! Please feel free to share any follow ups here, looking forward to further discussion to make this work robust!

---

> ### Author Response · Authors · 2025-11-28
>
> Since the discussion period is nearing its conclusion, we are writing to you with a gentle reminder to view and share your thoughts on our responses. If you have any additional questions or concerns, kindly let us know so we can run any additional experiments and resolve them before the discussion period concludes. If you feel our responses have satisfactorily addressed your concerns, we would be grateful for your reconsideration of this paper's score to show the same.
>
> We realize that this discussion period overlaps with the Thanksgiving holidays, so we want to express our gratitude for taking the time to share your thoughts here. Warm wishes for the upcoming holiday season.

---

### Author Response · Authors · 2025-12-03
**Summary for AC Review**

Given recent ICLR developments, to make life easier for upcoming reviewers, we are sharing a consolidated list of reviewer feedback and questions along with our responses and paper modifications.

**Tldr: CWIC is a new SOTA sparsity method using learnable thresholds, outperforms prior SOTA TEAL (ICLR Spotlight ‘25), and RSPARSE (ICLR ‘25) on quality, while matching TEAL’s triton kernel hardware acceleration gains.
The first sparsity method we know of where a sparsified larger model surpasses small dense models of the same family, with comparable active-parameters.**

We acknowledge our initial scores are lower due to requests to improve writing quality and concerns about (1) lack of comparison to R-Sparse, (2) training data choice and (3) missing ablations. We reworked the methods and results section, added a new baseline, detailed data used and added ablations. A comprehensive list of changes is in the table below.

**Paper Summary**

**Aim**: Construct a non-heuristic method to sparsify dense models to arbitrary sparsity levels while allowing variable compute allocation to different layers, tokens and sequences. The resulting model should outperform existing SOTA and result in realized inference speedups.

**Current SOTA**: TEAL (ICLR 2025, Spotlight) and R-SPARSE (ICLR 2025) replaces linear layers in LLMs with activation-aware column sparsity and rank-aware sparse inference respectively. These works are recent and have not been compared. We found that TEAL slightly outperforms RSPARSE.

**Result**: CWIC applies the bitter lessons to compute allocation and sparsity. Since CWIC makes sparsity thresholds learnable,  quality can be scaled with training compute. It designates different levels of sparsity to different inputs and layers. Finally, it significantly outperforms TEAL and R-SPARSE (ICLR 2025 contributions) across sparsity levels (50% to 84% sparse) when tested on Llama-3.1 1B and 3B models (see Results section). The dramatic performance improvement over existing methods is particularly pronounced at higher sparsity levels. We additionally provide kernel implementations that at 66% sparsity, boost inference by 2.5x on GPU and CPU.

**Observations**: CWIC allocates less compute to filler/replicated text and more to questions humans deem challenging. Among attention matrices, activation of the V matrix is the most dense, followed by K and O. Sparsity is greatest across matrices from layers 18-21 and 5-10 (therefore the other layers seem particularly important to model performance).

**Reviewer Discussion**

Overall, our reviewers understood our core motivation and posed mostly well-reasoned and thoughtful questions that have improved our paper quality. There were a few minor feedback points that were incorrect (likely due to swift reading of the paper) that we have addressed. Every writing suggestion, additional experiment and ablation request has been carefully considered and incorporated in the paper 2-3 days post review release. All clarification questions are addressed.

Unfortunately, reviewers 6kNm, dRcL and oPbm never got the chance to reply to our comment and revisit scores before the discussion lockdown. Reviewer AM9y felt that our “rebuttal answered most of [their] questions” and requested clarification on two more points before increasing scores which we have provided.

---

> ### Author Response · Authors · 2025-12-03
> **Steps taken to address feedback**
>
> | Feedback | How we addressed |
> |------------------|-----------------|
> | [Writing] A consistent request was to simplify notation/operators and re-organize the methods section for easier readability. | **We have reorganized and rewritten Methods, Section 3.** Notation has been simplified (and numbered). Overall, the section reads much better thanks in large part to the constructive criticism. |
> | [Experiments] The paper compares CWIC to TEAL. How does it compare to distillation and dense models? | Since there is no standard distillation recipe for arbitrary student model sizes, we compare to SOTA dense models at comparable parameter count (e.g., Gemma-3-270M, Llama-1B). **In Results, we now show that sparsifying with CWIC outperforms these dense models across benchmarks.** |
> | [Experiments] Are there other baselines to compare to besides TEAL? | In their 2025 paper, TEAL demonstrates outperformance of prior SOTA techniques such as CATS. A contemporary 2025 paper shows R-SPARSE also outperforms CATS and GRIFFIN. **On lighteval v0.10.0 we found TEAL to slightly outperform R-SPARSE. Both techniques collapse at higher sparsities and are outperformed by CWIC.** As a side note: R-SPARSE and TEAL have ceilings on sparsification since they do not modify the LMHead (likely due to collapse). CWIC does not face this ceiling as it sparsifies the LMHead as well. |
> | [Writing] Confusion on benchmark vs training data. | **Clarified in Methods and Appendix** that we use Fineweb for training. We use lighteval v0.10.0 to benchmark model performance. Neither the train nor test sets of the benchmarks (MMLU, ARC, HellaSwag, OpenBookQA, WinoGrande, PIQA) are used for training. |
> | [Code] Making codebase and kernel implementation accessible during review. | We planned to wait for publication (per the Reproducibility Statement) to preserve anonymity, but realized we could **anonymously release** the code at: https://anonymous.4open.science/r/anon-8BD5/. |
> | [Experiments] Seeking more clarity on hyperparameters used. | We have **provided a set of ablation studies** to demonstrate the impact of design decisions (full results in Appendix). All **hyperparameters are provided in an Appendix table**. Empirically, “stripe” size has the greatest impact. |
> | [Writing] What are the differences in training time of CWIC and TEAL? | At 50% sparsity, TEAL’s training-free evaluates 1M samples in <1 hour; CWIC takes ~3.5 days. CWIC takes longer to apply to a model than TEAL, but both are modest compared to methods like QSparse (which are used over the entirety of training time) or even training a smaller dense model from scratch at which point the significant performance gap makes CWIC favorable for inference quality. **We note this in Section 4.2.2**. |
> | [Writing] Results were partially presented in the introduction and results section. | All **figures and tables have been consolidated in the Results section**. We have also shown a breakdown of all aggregate results in the Appendix for completeness. |
> | [Experiments] Use of STE was not empirically justified. | We now **include this in the ablation section**. From an empirical point of view, our ablations without STE (inspired by SAE reconstruction) underperform ones that used STE. |
> | [Writing] The difference between CWIC and n:m structured sparsity was unclear. | **Clarified in “Related Methods”**: In CWIC, the effective sparse set of weights used differs per token per sequence. In n:m sparsity we make a one time reduction to the expressiveness and size of the weight matrix. |

---

> ### Author Response · Authors · 2025-12-03
> **Questions and clarifications**
>
> | Reviewer Question | Our Response |
> |------------------|-----------------|
> | [High-level] How does the method differ from DeepSeek’s Native Sparse Attention? | DeepSeek’s Native Sparse Attention is an attention sparsity mechanism, whereas CWIC is an activation sparsity method. NSA sparsifies the attention matrix, while CWIC reduces the number of non-zero activations in hidden states. |
> | [High-level] Are inference speedups compared to dense models? | Yes. We show speedup curves for both TEAL and CWIC, and we find that they match closely across sparsity levels. |
> | [High-level] How would this compare to pure distillation? | We faced two complications here. First, what model do we use as the student initialization? Second, how do we reach arbitrary FLOP reduction ratios? Since the research community hasn’t agreed on a standard distillation recipe, we decided to compare sparsified models to existing trained-from-scratch dense models at lower parameter count to enable extrapolating results. **This is provided in the Results section.** |
> | [High-level] How does batching work? | Since activations differ per time in the batch, it is challenging to share thresholds across a batch. We note that CWIC-like methods are intended for edge inference on consumer hardware where the batch size is often 1. At large batch sizes, there may be clever techniques that group similar activation sequences together. We consider this future work at this time. |
> | [Details] Why use normalization instead of setting RMSNorm scales to 1? | Setting the rms norm scales to 1 does not achieve our intended behavior since the student would no longer match the teacher at initialization. Also, RMSNorm uses RMS over the last hidden dimension across channels as the normalizing term for _all_ the channels. So relative disparity in channel scales would not be removed. |
> |[Details] Does post hoc threshold update affect differentiability? Can we parameterize the threshold directly in log space? | The update happens outside the autograd graph so it has no effect. We did consider log space but we found the current approach was cleaner since we start distillment with thresholds = 0 to match the teacher model. |
> | [Details] Were other relaxation methods tried? | We did try sigmoid and gumbel among many other designs and found that learning would degrade early on in sparsification. Given compute limits, we focused on the method that behaved well over full runs in favor of partial results from unstable early experiments. |

---

### Meta-Review · Area_Chair_tA71 · 2026-01-05

**Summary:**

Strength: The paper proposes a method for sparsity-aware inference in LLMs. The method is based on learnable and contextual sparsity threshold. Through experiments on Llama models, it demonstrates the effectiveness of the method.

Weakness: More baselines should be included for comparison. The presentation of the paper should be improved. Also, the scalability of the methods should be better demonstrated.

**Reviewer Concerns:**

Weakness: More baselines should be included for comparison. The presentation of the paper should be improved. Also, the scalability of the methods should be better demonstrated.

**Reviewer Scores:**

Based on the comments and disucssions, the reviewers will likely keep their scores.

---

### Decision · Program_Chairs · 2026-01-26

Reject